# Disentangling Extraction and Reasoning in Multi-hop Spatial Reasoning

**Roshanak Mirzaee**
Michigan State university
mirzaeem@msu.edu

**Parisa Kordjamshidi**
Michigan State university
kordjams@msu.edu

## Abstract

Spatial reasoning over text is challenging as the models not only need to extract the direct spatial information from the text but also reason over those and infer implicit spatial relations. Recent studies highlight the struggles even large language models encounter when it comes to performing spatial reasoning over text. In this paper, we explore the potential benefits of disentangling the processes of information extraction and reasoning in models to address this challenge. To explore this, we design various models that disentangle extraction and reasoning (either symbolic or neural) and compare them with state-of-the-art (SOTA) baselines with no explicit design for these parts. Our experimental results consistently demonstrate the efficacy of disentangling, showcasing its ability to enhance models' generalizability within realistic data domains.

## 1 Introduction

Despite the high performance of recent pretrained language models on question-answering (QA) tasks, solving questions that require multi-hop reasoning is still challenging (Mavi et al., 2022). In this paper, we focus on spatial reasoning over text which can be described as inferring the implicit[1] spatial relations from explicit relations[2] described in the text. Spatial reasoning plays a crucial role in diverse domains, including language grounding (Liu et al., 2022), navigation (Zhang et al., 2021), and human-robot interaction (Venkatesh et al., 2021). By studying this task, we can analyze both the reading comprehension and logical reasoning capabilities of models.

Previous work has investigated the use of general end-to-end deep neural models such as pretrained language models (PLM) (Mirzaee et al., 2021) in

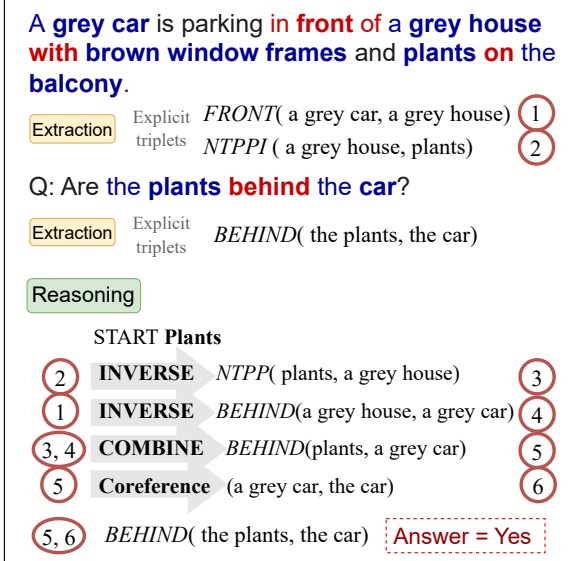

Figure 1: An example of steps of spatial reasoning on RESQ dataset. We begin by searching for *the plants* from the question triplet within the text, enabling us to extract explicit triplets (1,2). Next, we apply rules such as INVERSE to deduce implicit triplets (3,4,5). Then, utilizing triplets 5 and 6 we determine the final answer, 'Yes'. NTPP: Non-Tangential Proper Part (Table 1).

spatial question answering (SQA). PLMs show reasonable performance on the SQA problem and can implicitly learn spatial rules from a large set of training examples. However, the black-box nature of PLMs makes it unclear whether these models are making the abstractions necessary for spatial reasoning or their decisions are based solely on patterns observed in the data.

As a solution for better multi-hop reasoning, recent research has investigated the impact of using fine-grained information extraction modules such as Named Entity Recognition (NER) (Mollá et al., 2006; Mendes et al., 2010), gated Entity/Relation (Zheng and Kordjamshidi, 2021) or semantic role labels (SRL) (Shen and Lapata, 2007; Faghihi et al., 2023) on the performance of models.

---

[1]By implicit, we mean indirect relations, not metaphoric usages or implicit meaning for the relations.

[2]relationships between objects and entities in the environment, such as location, distance, and relative position.

| Formalism (General Type) | Spatial Type | Expressions (e.g.) |
|---|---|---|
| Topological (RCC8) | DC (disconnected)
EC (Externally Connected)
PO (Partially Overlapped)
EQ (Equal)
TPP
NTPP
TPPI
NTPPI | disjoint
touching
overlapped
equal
covered by
in, inside
covers
has |
| Directional (Relative) | LEFT, RIGHT
BELOW, ABOVE
BEHIND, FRONT | left of, right of
under, over
behind, in front |
| Distance | Far, Near | far, close |

Table 1: List of spatial relation formalism and types.

On a different thread, cognitive studies (Stenning and Van Lambalgen, 2012; Dietz et al., 2015) show when the given information is shorter, humans also find spatial abstraction and use spatial rules to infer implicit information. Figure 1 shows an example of such extractions. Building upon these findings, we aim to address the limitations of end-to-end models and capitalize on the advantages of fine-grained information extraction in solving SQA. Thus, we propose models which disentangle the *language understanding* and *spatial reasoning* computations as two separate components. Specifically, we first design a pipeline model that includes trained neural modules for extracting direct fine-grained spatial information from the text and performing symbolic spatial reasoning over them.

The second model is simply an end-to-end PLM that uses annotations used in extraction modules of pipeline model in the format of *extra QA* supervision. This model aims to demonstrate the advantages of using separate extraction modules compared to a QA-based approach while utilizing the same amount of supervision. Ultimately, the third model is an end-to-end PLM-based model on relation extraction tasks that has explicit latent layers to disentangle the extraction and reasoning inside the model. This model incorporates a neural spatial reasoner, which is trained to identify all spatial relations between each pair of entities.

We evaluate the proposed models on multiple SQA datasets, demonstrating the effectiveness of the disentangling extraction and reasoning approach in controlled and realistic environments. Our pipeline outperforms existing SOTA models by a significant margin on benchmarks with a controlled environment (toy tasks) while utilizing the same or fewer training data. However, in real-world scenarios with higher ambiguity of natural language for extraction and more rules to cover, our

end-to-end model with explicit layers for extraction and reasoning performs better.

These results show that disentangling extraction and reasoning benefits deterministic spatial reasoning and improves generalization in realistic domains despite the coverage limitations and sensitivity to noises in symbolic reasoning. These findings highlight the potential of leveraging language models for information extraction tasks and emphasize the importance of explicit reasoning modules rather than solely depending on black-box neural models for reasoning.

## 2    Related Research

**End-to-end model on SQA:** To solve SQA tasks, recent research evaluates the performance of different deep neural models such as Memory networks (Shi et al., 2022; Sukhbaatar et al., 2015), Self-attentive Associative Memory (Le et al., 2020), subsymbolic fully connected neural network (Zhu et al., 2022), and Recurrent Relational Network (RRN) (Palm et al., 2017). Mirzaee and Kordjamshidi; Mirzaee et al. use transfer learning and provide large synthetic supervision that enhances the performance of PLMs on spatial question answering. However, the results show a large gap between models and human performance on human-generated data. Besides, none of these models use explicit spatial semantics to solve the task. The only attempt towards integrating spatial semantics into spatial QA task is a baseline model introduced in (Mirzaee et al., 2021), which uses rule-based spatial semantics extraction for reasoning on bAbI (task 17) which achieves 100% accuracy without using any training data.

**Extraction and Reasoning:** While prior research has extensively explored the use of end-to-end models for learning the reasoning rules (Minervini et al., 2020; Qu et al., 2021), there is limited discussion on separating the extraction and reasoning tasks. Nye et al. utilizes LMs to generate new sentences and extract facts while using some symbolic rules to ensure consistency between generated sentences. Similarly, ThinkSum (Ozturkler et al., 2022) uses LMs for knowledge extraction (Think) and separate probabilistic reasoning (Sum), which sums the probabilities of the extracted information. However, none of these works are on multi-step or spatial Reasoning.

# 3 Proposed Models

To understand the effectiveness of disentangling the extraction and reasoning modules, we provide three groups of models. The first model is a pipeline of extraction and symbolic reasoning (§3.1), the second model is an end-to-end PLM that uses the same spatial information supervision but in a QA format (§3.2), and the third model is an end-to-end neural model with explicit layers of extraction and reasoning (§3.3). We elaborate each of these models in the subsequent sections.

**Task** The target task is spatial question answering (SQA), which assesses models' ability to comprehend spatial language and reason over it. Each example includes a textual story describing entities and their spatial relations, along with questions asking an *implicit* relation between entities (e.g., Figure 1). SQA benchmarks provide two types of questions: YN (Yes/No) queries about the existence of a relation between two groups of entities, and FR (Find Relation) seeks to identify all possible (direct/indirect) relations between them. The answer to these questions is chosen from a provided candidate list. For instance, the candidate list for FR questions can be a sublist of all relation types in Table 1.

## 3.1 Pipeline of Extraction and Reasoning

Here, we describe our suggested pipeline model designed for spatial question answering task, referred to as **PISTAQ**[3]. As shown in the extraction part of Figure 2, the spatial information is extracted first and forms a set of triplets for a story (Facts) and a question (Query). Then a coreference resolution module is used to connect these triplets to each other. Given the facts and queries, the spatial reasoner infers all implicit relations. The answer generator next conducts the final answer. Below we describe each module in more detail.

**Spatial Role Labeling (SPRL)** is the task of identifying and classifying the *spatial roles* of phrases within a text (including the Trajector, Landmark, and Spatial Indicator) and formalizing their *relations* (Kordjamshidi et al., 2010). Here, we use the same SPRL modules as in (Mirzaee and Kordjamshidi, 2022). This model first computes the token representation of a story and its question using a BERT model. Then a BIO tagging layer is applied on the tokens representations using (O, B-

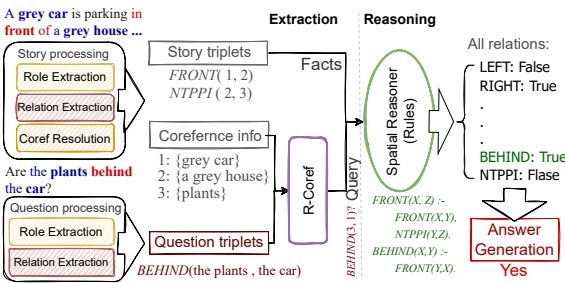

Figure 2: PISTAQ pipeline based on disentangled extraction and reasoning. In this model, facts, e.g., FRONT(grey car, grey house), are extracted from the story and linked by coreference modules. The R-Coref equates 'the car' from the question with 'a grey car' in the story and forms a query. This query, along with facts, is sent to the spatial reasoner. Finally, the spatial reasoner employs FRONT and BEHIND rules and returns True as the answer.

entity, I-entity, B-indicator, and I-indicator) tags. Finally, a softmax layer on the BIO tagger output selects the spatial entities[4] (e.g., 'grey car' or 'plants' in Figure 2) and spatial indicators (e.g., 'in front of' in Figure 2).

Given the output of the spatial role extraction module, for each combination of (Trajector, Spatial indicator, Landmark) in each sentence, we create a textual input[5] and pass it to a BERT model. To indicate the position of each spatial role in the sentence, we use segment embeddings and add 1 if it is a role position and 0 otherwise. The $[CLS]$ output of BERT will be passed to a one-layer MLP that provides the probability for each triplet. To apply the logical rules on the triplets, we need to assign a relation type to each triplet. To this aim, we use another multi-classification layer on the same $[CLS]$ token to identify the spatial types of the triplet. The classes are relation types in Table 1 alongside a class NaN for triplet with no spatial meaning. For instance, in Figure 2, (grey car, in front of, grey house) is a triplet with $FRONT$ as its relation type while (grey house, in front of, grey car) is not a triplet and its relation type is $NaN$. We use a joint loss function for triplet and relation type classification to train the model.

**Coreference Resolution** Linking the extracted triplets from the stories is another important step required in this task, as different phrases or pronouns may refer to same entity. To make such connections, we implement a coreference resolu-

---

[3]**PI**peline model for **S**pa**T**i**A**l **Q**uestion answering

[4]Trajector/Landmark

[5]$[CLS, traj, SEP, indic, SEP, land, SEP, sentence, SEP]$

| | | | | |
|---|---|---|---|---|
| Not | $\forall (X,Y) \in Entities$ | $R \in \{Dir \vee PP\}$ | IF $R(X,Y)$ | $\Rightarrow \mathrm{NOT}(R\_reverse(X,Y))$ |
| Inverse | $\forall (X,Y) \in Entities$ | $R \in \{Dir \vee PP\}$ | IF $R(Y,X)$ | $\Rightarrow R\_reverse(X,Y)$ |
| Symmetry | $\forall (X,Y) \in Entities$ | $R \in \{Dis \vee (RCC - PP)\}$ | IF $R(Y,X)$ | $\Rightarrow R(X,Y)$ |
| Transitivity | $\forall (X,Y,Z) \in Entities$ | $R \in \{Dir \vee PP\}$ | IF $R(X,Z), R(Z,Y)$ | $\Rightarrow R(X,Y)$ |
| Combination | $\forall (X,Y,Z,H) \in Entities$ | $R \in Dir, *PP \in PP$ | IF $*PP(X,Z), R(Z,H), *PPi(Z,Y)$ | $\Rightarrow R(X,Y)$ |

Table 2: Designed spatial rules (Mirzaee and Kordjamshidi, 2022). $Dir$: Directional relations (e.g., LEFT), $Dis$: Distance relations (e.g., FAR), $PP$: all Proper parts relations (NTPP, NTPPI, TPPI, TPP), $RCC - PP$: All RCC8 relation except proper parts relations. $*PP$: one of TPP or NTPP. $*PPi$: one of NTPPi or TPPi.

tion model based on (Lee et al., 2017) and extract all antecedents for each entity and assign a unique $id$ to them. In contrast to previous work, we have extended the model to support plural antecedents (e.g., two circles). More details about this model can be found in Appendix C.2. To find the mentions of the question entities in the story and create the queries, we use a Rule-based Coreference (R-Coref) based on exact/partial matching. In Figure 2, 'the car' in the question has the same id as 'the grey car' from the story's triplets.

**Logic-based Spatial Reasoner** To do symbolic spatial reasoning, we use the reasoner from (Mirzaee and Kordjamshidi, 2022). This reasoner is implemented in Prolog and utilizes a set of rules on various relation types, as illustrated in Table 2. Given the facts and queries in Prolog format, the spatial reasoner can carry out the reasoning process and provide an answer to any given query. The reasoner matches variables in the program with concrete values and a backtracking search to explore different possibilities for each rule until a solution is found. As shown in Figure 2, the reasoner uses a FRONT and a BEHIND rules over the facts and generates the True response for the query.

### 3.2 PLMs Using SPRL Annotations

To have a fair comparison between the QA baselines and models trained on SPRL supervision, we design **BERT-EQ**[6]. We convert the SPRL annotation into extra YN questions[7] asking about explicit relations between a pair of entities. To generate extra questions, we replace triplets from the SPRL annotation into the "Is [Trajector] [Relation*] [Landmark]?" template. The [Trajector] and [Landmark] are the entity phrases in the main sentence ignoring pronouns and general names (e.g., "an object/shape"). The [Relation*] is a relation expression (examples presented in Table 1) for the

triplet relation type. To have equal positive and negative questions, we reverse the relation in half of the questions. We train BERT-EQ using both original and extra questions by passing the "question+story" into a BERT with answers classification layers.

### 3.3 PLMs with Explicit Extractions

As another approach, we aim to explore a model that disentangles the extraction and reasoning parts inside a neural model. Here, rather than directly predicting the answer from the output of PLMs (as typically done in the QA task), we introduce explicit layers on top of PLM outputs. These layers are designed to generate representations for entities and pairs of entities, which are then passed to neural layers to identify all relations. We call this model **SREQA**[8], which is an end-to-end spatial relation extraction model designed for QA. Figure 3 illustrates the structure of this model.

In this model, we first select the entity mentions $(M_j(E_1))$ from the BERT tokens representation and pass it to the extraction part shown in Figure 3a. Next, the model computes entity representation $(M(E_1))$ by summing the BERT token representations of all entity's mentions and passing it to an MLP layer. Then for each pair of entities, a triplet is created by concatenating the pair's entities representations and the BERT $[CLS]$ token representation. This triplet is passed through an MLP layer to compute the final pair representations. Next, in the reasoning part in Figure 3a, for each relation type in Table 1, we use a binary 2-layer MLP classifier to predict the probability of each relation between the pairs. We remove the inconsistent relations by selecting one with a higher probability at inference time, e.g., LEFT and RIGHT cannot be true at the same time. The final output is a list of all possible relations for each pair. This model is trained using the summation of Focal loss (Lin et al., 2017) of all relation classifiers.

---

[6]BERT+**E**xtra **Q**uestion

[7]This augmentation does not apply to FR type since it inquires about all relations between the two asked entities.

[8]**S**patial **R**elation **E**xtraction for **QA**

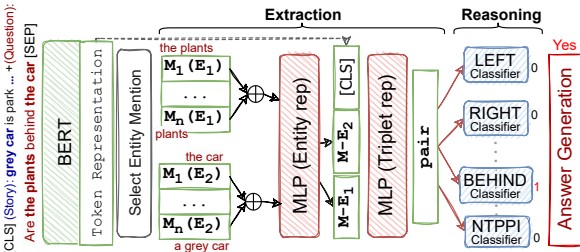

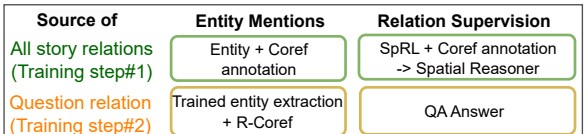

(a) Model structure. First, entity mentions such as 'plants' and 'grey car' are selected from the BERT output and the entity representation is formed. Next, triplets like ('plants', 'car', [CLS]) are generated and fed into the reasoning component. The collective output of all relation classifiers determines the relationships between each pair. *All hatched parts are trained end-to-end. The rest of the data is obtained from annotations or off-the-shelf modules.

| Source of | Entity Mentions | Relation Supervision |
|---|---|---|
| All story relations (Training step#1) | Entity + Coref annotation | SpRL + Coref annotation -> Spatial Reasoner |
| Question relation (Training step#2) | Trained entity extraction + R-Coref | QA Answer |

(b) The source of supervision in each step of training. In step#1, we train the model on all story relations, and in step#2, we only train it on question relations. These modules and data are the same as the ones used in PISTAQ.

Figure 3: The SREQA model with explicit neural layers to disentangle extraction and reasoning part.

We train SREQA in two separate steps. In the first step, the model is trained on a relation extraction task which extracts *all direct and indirect relations* between *each pair of entities* only from stories. The top row of Figure 3b shows the annotation and modules employed in this step to gather the necessary supervision. We use the entity and coreference annotation to select the entity mentions from the BERT output. To compute the relations supervision for each pair of entities, we employ the spatial reasoner from PISTAQ and apply it to the direct relations (triplets) from the SPRL annotation, which are connected to each other by coreference annotations. This step of training is only feasible for datasets with available SPRL and coreference annotations.

In the next step, we further train SREQA on extracting *questions relation* using QA supervision. As shown in the bottom row of Figure 3b, we employ the trained spatial role extraction model used in PISTAQ to identify the entities in the question and use R-Coref to find the mentions of these entities in the story. To obtain the relation supervision, we convert the question answers to relation labels. In FR, the label is similar to the actual answer, which is a list of all relations. In YN, the question

relation is converted to a label based on the Yes/No answer. For example, in Figure 3a, the question relation is 'BEHIND,' and the answer is Yes, so the label for the BEHIND classifier is 1.

We evaluate the SREQA model's performance in predicting the accurate answers of the test set's questions same as training step 2.

## 4 Experiments

### 4.1 Datasets

**SPARTQA** is an SQA dataset in which examples contain a story and multiple YN[9] and FR questions that require multi-hop spatial reasoning to be answered. The stories in this dataset describe relations between entities in a controlled (toy task) environment. This dataset contains a large synthesized part, SPARTQA-AUTO, and a more complex small human-generated subset, SPARTQA-HUMAN. All stories and questions in this dataset also contain SPRL annotations.

**SPARTUN** is an extension of SPARTQA with YN and FR questions containing SPRL annotations. Compared to SPARTQA, the vocabulary and relation types in this dataset are extended, and it covers more relation types, rules, and spatial expressions to describe relations between entities.

**RESQ** is an SQA dataset with Yes/No questions over the human-generated text describing spatial relations in real-world settings. The texts of this dataset are collected from **MSPRL** dataset (Kordjamshidi et al., 2017) (See Figure 1), which describe some spatial relations in pictures of ImageCLEF (Grubinger et al., 2006). Also, the MSPRL dataset already contains some SPRL annotations. To answer some of the questions in this dataset, extra spatial-commonsense information is needed (e.g., a roof is on the *top* of buildings).

### 4.2 Model Configurations & Baselines

We compare the models described in section 3 with the following baselines.

**Majority Baseline**: This baseline selects the most frequent answer(s) in each dataset.

**GT-PISTAQ:** This model uses ground truth (GT) values of all involved modules in PISTAQ to eliminate the effect of error propagation in the pipeline. This baseline is used to evaluate the alignments between the questions and story entities and the reasoning module in solving the QA task. It also

---

[9] We ignore "Dont know" answers in YN question and change them to No

| Model | Supervisions | Rule-based Modules |
|---|---|---|
| BERT | QA | - |
| GPT3.5$^{zero\_shot}$ | - | - |
| GPT3.5$^{few\_shot}$ | QA(8 ex) | - |
| GPT3.5$^{few\_shot}$+CoT | QA(8 ex) + CoT | - |
| BERT-EQ | QA +SpRL(S) | - |
| SREQA | QA +SpRL(all)+Coref | Reasoner, R-Coref |
| SREQA* | QA + SpRL(Q) | R-Coref |
| PISTAQ | SpRL(all) + Coref | Reasoner, R-Coref |
| PISTAQ$^{zero\_shot}$ | - | Reasoner, R-Coref |

Table 3: The list of annotations from the target benchmarks and rule-based modules employed in each model. We use a quarter of SPRL annotations to train the modules on auto-generated benchmarks. S: Stories, Q: Questions, All: Stories+Questions.

gives an upper bound for the performance of the pipeline model, as the extraction part is perfect.

**BERT**: We select BERT as a candidate PLM that entangles the extraction and reasoning steps. In this model, the input of the "question+story" is passed to the BERT, and the $[CLS]$ representation is used to do the answer classification.

**GPT3.5**: GPT3.5 (Brown et al., 2020) baselines (GPT3.5 text-davinci-003) is selected as a candidate of generative larger language models which already passes many SOTAs in reasoning tasks (Bang et al., 2023; Kojima et al., 2022). We use $Zero\_shot$ and $Few\_shot$ (In-context learning with few examples) settings to evaluate this model on the human-generated benchmarks. We also evaluate the Chain-of-Thoughts (CoT) prompting method (Wei et al., 2022) to extend the prompts with manually-written reasoning steps. The format of the input and some prompt examples are presented in Appendix E.

Transfer learning has already demonstrated significant enhancements in numerous deep learning tasks (Soroushmojdehi et al., 2022; Rajaby Faghihi and Kordjamshidi, 2021). Thus, when applicable, we further train models on SPARTUN synthetic data shown by "*". The datasets' examples and statistics and more details of the experimental setups and configurations are provided in Appendix A and B. All codes are publicly available at https://github.com/RshNk73/PistaQ-SREQA.

## 5 Results and Discussion

Here, we discuss the influence of disentangling extraction and reasoning manifested in PISTAQ and SREQA models compared to various end-to-end models with no explicit design for these modules, such as BERT, BERT-EQ, and GPT3.5. Table 3 shows the list of these models with the sources of

their supervision as well as extra off-the-shelf or rule-based modules employed in them.

Since the performance of extraction modules, Spatial Role Labeling (SPRL) and Coreference Resolution (Coref), directly contribute to the final accuracy of the designed models, we have evaluated these modules and reported the results in Table 4. We choose the best modules on each dataset for experiments. For a detailed discussion on the performance of these modules, see Appendix C.

| Dataset | Coref | SRole | SRel | SType |
|---|---|---|---|---|
| MSPRL | - | 88.59 | 69.12 | 19.79 |
| MSPRL* | - | 88.03 | 71.23 | 23.65 |
| HUMAN | 82.16 | 55.8 | S: 57.43 | 43.79 |
| | | | Q: 52.55 | 39.34 |
| HUMAN* | 81.51 | 72.53 | S: 60.24 | 48.74 |
| | | | Q: 61.53 | 48.07 |
| SPARTQA | 99.83 | 99.92 | S: 99.72 | 99.05 |
| | | | Q: 98.36 | 98.62 |
| SPARTUN | 99.35 | 99.96 | S: 99.18 | 98.57 |
| | | | Q: 97.68 | 98.11 |

Table 4: Performance of the extraction modules. Q: question. S: stories. HUMAN: SPARTQA-HUMAN. SPARTQA: SPARTQA-AUTO. *Further pretraining modules on SPARTUN. We report macro F1 for SPRL and the accuracy of the Coref modules.

### 5.1 Result on Controlled Environment

Table 5 shows the performance of models on two auto-generated benchmarks, SPARTUN and SPARTQA-AUTO. We can observe that PISTAQ outperforms all PLM baselines and SREQA. This outcome first highlights the effectiveness of the extraction and symbolic reasoning pipeline compared to PLMs in addressing deterministic reasoning within a controlled environment. Second, it shows that disentangling extraction and reasoning as a pipeline works better than explicit neural layers in SQA with a controlled environment. The complexity of these environments is more related to conducting several reasoning steps and demands accurate logical computations where a rule-based reasoner excels. Thus, the result of PISTAQ with a rule-based reasoner module is also higher than SREQA with a neural reasoner.

The superior performance of PISTAQ over BERT suggests that SPRL annotations are more effective in the PISTAQ pipeline than when utilized in BERT-EQ in the form of QA supervision. Note that the extraction modules of PISTAQ achieve perfect results on auto-generated benchmarks while trained only on a quarter of the SPRL annotations

| # | Models | SPARTUN | | SPARTQA-AUTO | |
|---|---|---|---|---|---|
| | | YN | FR | YN | FR |
| 1 | Majority baseline | 53.62 | 14.23 | 51.82 | 44.35 |
| 2 | GT-PISTAQ | 99.07 | 99.43 | 99.51 | 98.99 |
| 3 | BERT | 91.80 | 91.80 | 84.88 | 94.17 |
| 4 | BERT-EQ | 90.71 | N/A | 85.60 | N/A |
| 5 | SREQA | 88.21 | 83.31 | 85.11 | 86.88 |
| 6 | PISTAQ | **96.37** | **94.52** | **97.56** | **98.02** |

Table 5: Results on auto-generated datasets. We use the accuracy metric for both YN and FR questions.

as shown in Table 5. However, BERT-EQ uses all the original dataset questions and extra questions created from the full SPRL annotations.

Table 6 demonstrates the results of models on SPARTQA-HUMAN with a controlled environment setting. As can be seen, our proposed pipeline, PISTAQ, outperforms the PLMs by a margin of 15% on YN questions, even though the extraction modules, shown in Table 4, perform low. This low performance is due to the ambiguity of human language and smaller training data. We also evaluate PISTAQ on SPARTQA-HUMAN FR questions using Macro_f1 score on all relation types. PISTAQ outperforms all other baselines on FR questions, except for BERT*.

There are two main reasons behind the inconsistency in performance between YN and FR question types. The first reason is the complexity of the YN questions, which goes beyond the basics of spatial reasoning and is due to using quantifiers (e.g., all circles, any object). While previous studies have demonstrated that PLMs struggle with quantifiers (Mirzaee et al., 2021), the reasoning module in PISTAQ can adeptly handle them without any performance loss. Second, further analysis indicates that PISTAQ predicts 'No' when a relationship is not extracted, which can be correct when the answer is 'No'. However, in FR, a missed extraction causes a false negative which decreases F1 score.

## 5.2 Results on Real-world Setting

We select RESQ as an SQA dataset with realistic settings and present the result of models on this dataset in Table 7.

To evaluate PISTAQ on RESQ, we begin by adapting its extraction modules through training on the corresponding dataset. We train the SPRL modules on both MSPRL and SPARTUN, and the performance of these models is presented in Table 4. As the MSPRL dataset lacks coreference an-

| # | Models | YN | FR | | |
|---|---|---|---|---|---|
| | | Acc | P | R | F1 |
| 1 | Majority baseline | 52.44 | 29.87 | 14.28 | 6.57 |
| 2 | GT-PISTAQ | 79.72 | 96.38 | 66.04 | 75.16 |
| 3 | BERT | 51.74 | 30.74 | 30.13 | 28.17 |
| 4 | BERT* | 48.95 | 60.96 | 49.10 | **50.56** |
| 5 | GPT3.5$^{Zero\_shot}$ | 45.45 | 40.13 | 22.42 | 16.51 |
| 6 | GPT3.5$^{Few\_shot}$ | 60.13 | 45.20 | 54.10 | 44.28 |
| 7 | GPT3.5$^{Few\_shot}$+CoT | 62.93 | 57.18 | 37.92 | 38.47 |
| 8 | BERT-EQ | 50.34 | - | - | - |
| 9 | BERT-EQ* | 45.45 | - | - | - |
| 10 | SREQA | 53.23 | 15.68 | 13.85 | 13.70 |
| 11 | SREQA* | 46.96 | 18.70 | 25.79 | 24.61 |
| 12 | PISTAQ | **75.52** | 72.11 | 35.93 | 46.80 |

Table 6: Results on SPARTQA-HUMAN. We use accuracy on YN questions and average Precision (P), Recall (R), and Macro-F1 on FR question types. *Using SPARTUN supervision for further training.

| # | Models | Accuracy |
|---|---|---|
| 1 | Majority baseline | 50.21 |
| 2 | BERT | 57.37 |
| 3 | BERT*$^{Zero\_shot}$ | 49.18 |
| 4 | BERT* | 63.60 |
| 5 | GPT3.5$^{Zero\_shot}$ | 60.32 |
| 6 | GPT3.5$^{Few\_shot}$ | 65.90 |
| 7 | GPT3.5$^{Few\_shot}$+CoT | 67.05 |
| 8 | BERT-EQ | 56.55 |
| 9 | BERT-EQ*$^{Zero\_shot}$ | 51.96 |
| 10 | BERT-EQ* | 61.47 |
| 11 | SREQA | 53.15 |
| 12 | SREQA*$^{Zero\_shot}$ | 53.32 |
| 13 | SREQA* | **69.50** |
| 14 | PISTAQ$^{MSPRL}$ | 41.96 |
| 15 | PISTAQ $^{SPARTUN+MSPRL}$ | 47.21 |
| 16 | Human | 90.38 |

Table 7: Result on RESQ. *Further training on SPARTUN. The $Zero\_shot$ refers to evaluation without further training on RESQ or MSPRL training data.

notations, we employ the model trained on SPARTUN for this purpose. Rows 14 and 15 in Table 7 show the performance of the PISTAQ on RESQ is inferior compared to other baselines. To find the reason, we analyze the first 25 questions from the RESQ test set. We find that 18 out of 25 questions required spatial commonsense information and cannot be answered solely based on the given relations in the stories. From the remaining 7 questions, only 2 can be answered using the SPRL annotations provided in the MSPRL dataset. Some examples of this analysis are provided in Appendix D. Hence, the low performance of PISTAQ is attributed to first the absence of integrating commonsense information in this model and, second, the errors in the

| | |
|---|---|
| **SparTQA-Human** | *Story:* There is a block called A. Inside there's a red oval that is above a blue triangle ... There is a block called B which is to the right and touching A. Inside B there are 5 triangles, a red one, a green one, a large yellow one, a brown one, and an orange one ... The large yellow triangle is touching the top edge of B. The green triangle is touching the right edge of B... |
| | *Question 1:* Is the green thing above the yellow thing? Answer: No |
| | *Question 2:* Does B have all of the triangles? Answer: No |
| **GPT3.5 (Prediction)** | Q1:The green triangle is to the right of the large yellow triangle and the large yellow triangle is touching the top edge of B. Using inverse rule, the green triangle is below the yellow triangle. The answer is **No**. |
| | Q2: There are five triangles mentioned in block B, a red one, a green one, a large yellow one, a brown one, and an orange one. All of these triangles are in block B, so the answer is **Yes**. |
| **GPT4 (Prediction)** | Q1: The green triangle is touching the right edge of B, and the large yellow triangle is touching the top edge of B. It is not mentioned if the green triangle is above or below the yellow triangle. The answer is **Don't Know**. |
| | Q2: There is a blue triangle in block A, and there are 5 triangles in block B. Since there is a triangle in block A, not all triangles are in block B. The answer is **No**. |
| **PaLM2 (Prediction)** | Q1: The green triangle is touching the right edge of B and the large yellow triangle is touching the top edge of B. The right edge of B is below the top edge of B. So the green triangle is below the yellow triangle. The answer is **No**. |
| | Q2: There are 5 triangles in B. There is a red triangle in B. There is a green triangle in B. There is a large yellow triangle in B. There is a brown triangle in B. There is an orange triangle in B. The answer is Yes. |
| **PISTAQ** | Q2: Extracted Facts (GPT3, Turbo):    NTPP(blue triangle, A), TPP(red triangle, left edge of B), TPP(orange triangle, left edge of B), ...
Extracted Queries (GPT3, Turbo):    NTPPI/TPPI (B, blue triangle) , NTPPI/TPPI (B, red triangle) , NTPPI/TPPI (B, orange triangle) , ...
Symbolic Reasoning (Rule-based Reasoner):   False & True & True & ... → Answer = **No** |

Figure 4: An example of Large Language Models (LLMs) prediction on SQA task evaluated with CoT prompting. The last row shows an example of using GPT3.5-Turbo for information extraction in PISTAQ. See Appendix E for *zero_shot* examples.

extraction modules, which are propagated to the reasoning modules.

As shown in Table 7, the best result on RESQ is achieved by SREQA* model. Compared to SREQA, SREQA* is trained on SPARTUN instead of MSPRL[10] in the first step of the training. MSPRL lacks some SPRL and coreference annotations to answer RESQ questions. In the absence of this information, collecting the supervision for the first phase of training results in a significant number of missed relations. Therefore, as shown in row 11 of Table 7, employing MSPRL in the first training phase decreases the performance while replacing it with SPARTUN in SREQA* significantly enhances the results.

SREQA* surpasses the PLMs trained on QA and QA+SPRL annotation, showcasing the advantage of the design of this model in utilizing QA and SPRL data within explicit extraction layers and the data preprocessing. Also, the better performance of this model compared to PISTAQ demonstrates how the end-to-end structure of SREQA can handle the errors from the extraction part and also can capture some rules and commonsense knowledge from RESQ training data that are not explicitly supported in the symbolic reasoner.

In *conclusion*, compared to PLMs, disentangling extraction and reasoning as a pipeline indicates su-

perior performance in deterministic spatial reasoning within controlled settings. Moreover, explicitly training the extraction module proves advantageous in leveraging SPRL annotation more effectively compared to using this annotation in QA format in the end-to-end training. Comparison between disentangling extraction and reasoning as a pipeline and incorporating them within an end-to-end model demonstrates that the end-to-end model performs better in realistic domains even better than PLMs. The end-to-end architecture of this model effectively enhances the generalization in the real-world setting and addresses some of the limitations of rule coverage and commonsense knowledge.

### 5.3 LLMs on Spatial Reasoning

Recent research shows the high performance of LLMs with *zero/few_shot* setting on many tasks (Chowdhery et al., 2022; Brown et al., 2020). However, (Bang et al., 2023) shows that Chat-GPT (GPT3.5-Turbo) with *zero_shot* evaluation cannot perform well on SQA task using SPARTQA-HUMAN test cases. Similarly, our experiments, as shown in Tables 6 and 7, show the lower performance of GPT3.5 (davinci) with *zero/few_shot* settings compared to human and our models PIS-TAQ and SREQA. Figure 4, shows an example of three LLMs, GPT3.5, GPT4 and PaLM2 on SPARTQA-HUMAN example[11] (complete figure

---

[10]As mentioned, we use the MSPRL annotation for RESQ dataset.

[11]Due to the limited resources, we only use GPT4 and PaLM2 on a few examples to evaluate their performance on

| Story: | a photo of a room with white walls , **two single beds** with a night table in between and **a picture** on the wall **above the beds** . |
|---|---|
| Question: | Are the beds below the picture? Answer: Yes |
| Story Facts: | BERT: 0: ['a picture', 'the beds'], 2:['a'], 1: ['a picture', 'the wall'] Facts: right(2, 1), below(2, 0), near(2, 0) |
| | GPT3: 3: ['two single beds', 'the beds'], 5: ['a picture'], 6: ['the wall', 'the beds'] Facts: above(5, 3), above(5, 6) ... |
| Queries: | BERT: below(0 , 0)? or below(0 , 1)? |
| | GPT3: below(3 , 5)? or below(3 , 6)? |
| Reasoning: | BERT: below(0 , 0) = False, below(0 , 1) = False  →  Answer = No |
| | GPT3: below(3 , 5) = True, below(3 , 6) = False → Answer = Yes |

Figure 5: An example of using BERT-based SPRL and GPT3.5 as information extraction in PISTAQ on a RESQ example.

including $zero\_shot$ examples is presented in Appendix E). Although Wei et al. shows that using CoT prompting improves the performance of PaLM on multi-step reasoning task, its spatial reasoning capabilities still does not meet the expectation.

### 5.3.1 LLMs as Extraction Module in PISTAQ

A recent study (Shen et al., 2023) shows that LLMs have a promising performance in information retrieval. Building upon this, we employ LLM, GPT3.5-Turbo with $few\_shot$ prompting to extract information from a set of SPARTQA-HUMAN and RESQ examples that do not necessitate commonsense reasoning for answering. The extracted information is subsequently utilized within the framework of PISTAQ.

The results, illustrated in the last row of Figure 4, highlight how the combination of LLM extraction and symbolic reasoning enables answering questions that LLMs struggle to address. Furthermore, Figure 5 provides a comparison between the trained BERT-based SPRL extraction modules and GPT3.5 with $few\_shot$ prompting in PISTAQ. It is evident that GPT3.5 extracts more accurate information, leading to correct answers. As we mentioned before, out of 25 sampled questions from RESQ, only 7 can be solved without relying on spatial commonsense information. Our experimental result shows that PISTAQ using LLM as extraction modules can solve all of these 7 questions.

Based on these findings, leveraging LLMs in PISTAQ to mitigate errors stemming from the SPRL extraction modules rather than relying solely on LLMs for reasoning can be an interesting future research direction. This insight emphasizes the importance of considering new approaches for incorporating explicit reasoning modules whenever possible instead of counting solely on black-box SQA tasks.

neural models for reasoning tasks.

## 6   Conclusion and Future Works

We investigate the benefits of disentangling the processes of extracting spatial information and reasoning over them. To this end, we devised a series of experiments utilizing PLMs for spatial information extraction coupled with a symbolic reasoner for inferring indirect relations. The outcomes of our experiments provide noteworthy insights: (1) Our observations in controlled experimental conditions demonstrate that disentangling extraction and symbolic reasoning, compared to PLMs, enhances the models' reasoning capabilities, even with comparable or reduced supervision. (2) Despite the acknowledged fragility of symbolic reasoning in real-world domains, our experiments highlight that employing explicit extraction layers and utilizing the same symbolic reasoner in data preprocessing enhances the reasoning capabilities of models. (3) Despite the limitations of LLMs in spatial reasoning, harnessing their potential for information extraction within a disentangled structure of Extraction and Reasoning can yield significant benefits. All of these results emphasize the advantage of disentangling the extraction and reasoning in spatial language understanding.

In future research, an intriguing direction is incorporating spatial commonsense knowledge using LLMs as an extraction module in the pipeline of extraction and reasoning. Additionally, the model's applicability extends beyond spatial reasoning, making it suitable for various reasoning tasks involving logical rules, such as temporal or arithmetic reasoning.

### Acknowledgements

This project is partially supported by the National Science Foundation (NSF) CAREER award 202826. Any opinions, findings, and conclusions or recommendations expressed in this material are those of the authors and do not necessarily reflect the views of the National Science Foundation. We thank all reviewers for their helpful comments and suggestions. We would like to express our gratitude to Hossein Rajaby Faghihi for his valuable discussions and input to this paper.

### Limitations

Our model is evaluated on a Spatial Reasoning task using specifically designed spatial logical rules.

However, this methodology can be readily extended to other reasoning tasks that involve a limited set of logical rules, which can be implemented using logic programming techniques. The extraction modules provided in this paper are task-specific and do not perform well on other domains, but they can be fine-tuned on other tasks easily. Using LLM in the extraction phase can also deal with this issue. Also, using MSPRL annotation on RESQ(which this data is provided on) decreases the performance of our models. This annotation does not contain the whole existing relations in the context. The evaluation of the reasoning module is based on the existing datasets. However, we cannot guarantee that they cover all possible combinations between spatial rules and relation types. Many questions in RESQ need spatial commonsense to be answered. As a result, due to the limitation of our symbolic spatial reasoner, the performance of the pipeline model is much lower than what we expected. Due to the high cost of training GPT3.5 on large synthetic data, we cannot fine-tune the whole GPT3.5 and only provide the GPT3.5 with $Few\_shot$ learning on small human-generated benchmarks. Also, due to the limited access, we can only test PaLM2 and GPT4 on a few examples.

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

# A Statistic Information

This section presents statistical information regarding dataset sizes and additional analyses conducted on the evaluation sets of human-generated datasets.

Table 8 provides the number of questions in the training and evaluation sets of the SQA benchmarks. Tables 9 and 10 present the sentence and number of relation triplets within the SPRL annotation for each dataset, respectively. Table 11 illustrates a comprehensive breakdown of the size of Role and Relation sets in the MSPRL dataset.

| Dataset | Train | Dev | Test |
|---|---|---|---|
| SPARTQA-AUTO (YN) | 26152 | 3860 | 3896 |
| SPARTQA-AUTO (FR) | 25744 | 3780 | 3797 |
| SPARTQA-HUMAN (YN) | 162 | 51 | 143 |
| SPARTQA-HUMAN (FR) | 149 | 28 | 77 |
| SPARTUN (YN) | 20334 | 3152 | 3193 |
| SPARTUN (FR) | 18400 | 2818 | 2830 |
| RESQ(YN) | 1008 | 333 | 610 |

Table 8: Number of questions in training and evaluation sets of SQA benchmarks.

| Dataset | Train | Dev | Test |
|---|---|---|---|
| SPARTQA-AUTO (story) | 36420 | 16214 | 16336 |
| SPARTQA-AUTO (question) | 53488 | 15092 | 15216 |
| SPARTQA-HUMAN (story) | 389 | 213 | 584 |
| SPARTQA-HUMAN (question) | 623 | 190 | 549 |
| SPARTUN (story) | 68048 | 9720 | 10013 |
| SPARTUN (question) | 41177 | 6355 | 6340 |
| MSPRL | 600 | - | 613 |

Table 9: Number of sentences in SPRL annotations of each benchmarks. To train models on the auto-generated benchmarks, we only use the quarter of training examples from SPARTUN and SPARTQA-AUTO.

| Dataset | Train | Dev | Test |
|---|---|---|---|
| SPARTQA-AUTO (story) | 159712 | 22029 | 21957 |
| SPARTQA-AUTO (question) | 232187 | 34903 | 35011 |
| SPARTQA-HUMAN (story) | 176 | 99 | 272 |
| SPARTQA-HUMAN (question) | 155 | 127 | 367 |
| SPARTUN (story) | 48368 | 7031 | 7191 |
| SPARTUN (question) | 38734 | 5970 | 6023 |
| MSPRL | 761 | - | 939 |

Table 10: Number of triplets in SPRL annotations of each benchmarks.

## A.1 Analyzing SPARTQA-HUMAN YN

We conducted additional evaluations on the superior performance of PISTAQ over other baseline models on SPARTQA-HUMAN YN questions. As explained before, PISTAQ tends to predict *No* when

|  | Train | Test | All |
|---|---|---|---|
| Sentences | 600 | 613 | 1213 |
| Trajectors | 716 | 874 | 1590 |
| Landmarks | 612 | 573 | 1185 |
| Spatial Indicators | 666 | 795 | 1461 |
| Spatial Triplets | 761 | 939 | 1700 |

Table 11: MSPRL size (Kordjamshidi et al., 2017).

---

Three boxes called one, two and three exist in an image. Box one contains a big yellow melon and a small orange watermelon. **Box two has a small yellow apple**. A small orange apple is inside and touching this box. Box one is in box three. **Box two** is to the **south** of, **far from** and to the **west** of **box three**. A **small yellow watermelon** is **inside box three**.

Q: Is **the yellow apple** to the **west** of the **yellow watermelon**? Yes

Q: Where is **box two** relative to the **yellow watermelon**? Left, Below, Far

---

Figure 6: An example of SPARTUN dataset from (Mirzaee and Kordjamshidi, 2022).

information is not available, resulting in more *No* and fewer *Yes* predictions compared to other models, as presented in Table 12. The number of true positive predictions for PISTAQ is more than two other baselines, and as a result, it achieves higher accuracy.

| Predictions/ Answer | Yes | No | No prediction |
|---|---|---|---|
| Ground Truth | 74 | 69 | - |
| BERT | 131 | 12 | - |
| BERT* | 89 | 54 | - |
| PISTAQ | 43 | 97 | 3 |

Table 12: Detailed information about the prediction of PISTAQ and BERT on SPARTQA-HUMAN YN questions. "No prediction" is related to the PISTAQ model when no correct SPRL extraction was made for the text of the question, and as a result, we have no answer prediction.

## A.2 SPRL Annotations in MSPRL

RESQ is built on the human-written context of MSPRL dataset which includes SPRL annotations. However, using this annotation in BERT-EQ and SREQA models causes lower performance (Check the result on Tablel 14). Our analysis shows that the

**Story:** behind it a bar with chairs and two people , and a bench with one person lying on it . Upper level with doors and a blue rail.

**mSpRL annotation:**   Triplet:  Behind (a bar (id: t1), behind, it (id: l1))
                                  Triplet:  Behind (a bench (id: t2), behind, it (id: l1))
                                  Triplet:  EC (one person (id: t3), on, it (id: l2))

**Question 1:**  Are the people behind the bar?     **Answer:**  Yes
**Predicted answer** based on mSpRL annotations:  **No**

**Question 2:**  Is the door above the bar?     **Answer:**  Yes
**Predicted answer** based on relation in text:  **No**
**Predicted answer** based on the commonsense (upper level is above the main level): **Yes**

Figure 7: An example of the limitation of MSPRL and coreference annotation to answer RESQ question. The answer of the questions predicted wrongly due to two main reasons. First, the missed commonsense knowledge in question 2 and second, the limited coverage of ground truth annotation in MSPRL in question 2.

SPRL annotations of MSPRL are not fully practical in our work due to two main reasons:

1. **Missed annotations:** As shown in Figure 7, there are many missed annotations for each text, e.g., NTPP(bar, with, chair).

2. **No coreference :** The coreference is not supported in this dataset, e.g., "L2: it" and "T2: a bench" are the same entity with different mentions, but they are mentioned with different ids. These missed coreferences result in fewer connections between entities and fewer inferred relations.

## B   Models and modules configuration

We use the huggingFace[12] implementation of pre-trained BERT base models, which have 768 hidden dimensions. All models are trained on the training set, evaluated on the dev set, and reported the result on the test set. For training, we train the model until no changes happen on the dev set and then store and use the best model on the dev set. We use AdamW ((Loshchilov and Hutter, 2017)), and learning rates from $2 \times 10^{-6}$, $2 \times 10^{-5}$ (depends on the task and datasets) on all models and modules. For the extraction modules and the baselines, we used the same configuration and setting as previous works (Mirzaee and Kordjamshidi, 2022). For SREQA models we use learning rates of $2 \times 10^{-5}$, $4 \times 10^{-6}$ for SREQA(story) and SREQA(question) respectively. To run the models we use machine with Intel Core i9-9820X (10 cores,

[12]https://huggingface.co/transformers/v2.9.1/model_doc/bert.html

3.30 GHz) CPU and Titan RTX with NVLink as GPU.

For GPT3.5, we use Instruct-GPT, *davinci-003*[13]. The cost for running GPT3.5 on the human-generated benchmarks was 0.002$ per 1k tokens. For GPT3.5 as information extraction, we use *GPT3.5 turbo* (a.k.a ChatGPT) with a cost of 0.0001$ per 1k tokens. We also use the GPT4 playground in OpenAI and PaLM2 playground to find the prediction of examples in Figure 11.

## C   Extraction and Reasoning Modules

Here, we discuss each module used in PISTAQ and their performance including the *Spatial Role Labeling* (SPRL), *Coreference Resolution*, and *Spatial reasoner*.

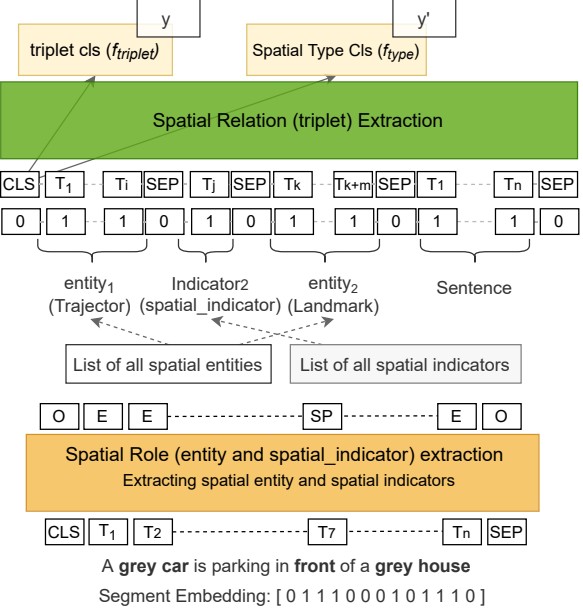

Figure 8: Spatial role labeling model includes two separately trained modules. E: entity, SP: spatial_indicators. As an example, triplet (a grey house, front , A grey car) is correct and the "spatial_type = FRONT", and (A grey car, front, a grey house) is incorrect, and the "spatial_type = NaN". Image from (Mirzaee and Kordjamshidi, 2022)

### C.1   Spatial Role Labeling (SPRL)

The SPRL module, shown in Figure 8 is divided into three sub-modules, namely, spatial role extraction (SRole), spatial relation extraction (SRel)[14], and spatial type classification (SType). We only

[13]from https://beta.openai.com
[14]Since the questions(Q) and stories(S) have different annotations (questions have missing roles), we separately train and test the SRel and SType modules

use these modules on sentences that convey spatial information in each benchmark. To measure the performance of SPRL modules, we use the macro average of F1 measure for each label. These modules are evaluated on three datasets that provide SPRL annotations, MSPRL, SPARTQA, and SPARTUN. When training the SPRL module on auto-generated benchmarks, we achieved a performance of 100% using only a quarter of the training data, therefore we stopped further training.

As shown in Table 4, all SPRL sub-modules achieve a high performance on synthetic datasets, SPARTQA and SPARTUN. This good performance is because these datasets may contain less ambiguity in the natural language expressions. Therefore, the BERT-base models can easily capture the syntactic patterns needed for extracting the roles and direct relations from the large training set.

## C.2  Coreference Resolution (Coref) in Spatial Reasoning

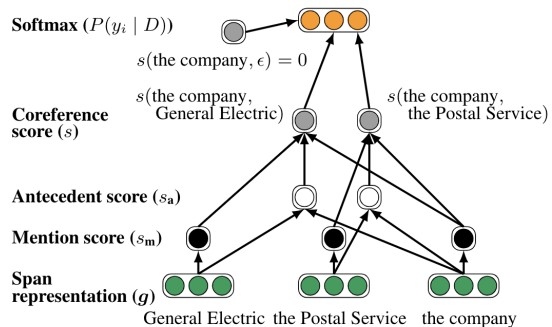

(a) The coreference resolution model structure.

$$s_{\mathrm{m}}(i) = \boldsymbol{w}_{\mathrm{m}} \cdot \mathrm{FFNN}_{\mathrm{m}}(\boldsymbol{g}_i)$$
$$s_{\mathrm{a}}(i,j) = \boldsymbol{w}_{\mathrm{a}} \cdot \mathrm{FFNN}_{\mathrm{a}}([\boldsymbol{g}_i, \boldsymbol{g}_j, \boldsymbol{g}_i \circ \boldsymbol{g}_j, \phi(i,j)])$$

(b) The formula for computing the coreference scores

Figure 9: The coreference resolution model (Lee et al., 2017).

We implement a coreference resolution model based on (Lee et al., 2017) to extract all antecedents for each entity (check Figure 9a). Compared to the previous works, we have the entities (phrase) annotations. Hence, we ignore the phrase scoring modules and use this annotation instead. We first collect all mentions of each predicted entity from spatial role extraction or role annotations, then assign an "id" to the same mentions and include that id in each triplet. For example, for BELOW(a cat,

| Datasets | Q-TYPE | Total | A | C | R |
|---|---|---|---|---|---|
| SPARTQA-AUTO | YN | 18 | 7 | 10 | 1 |
| | FR | 38 | 5 | 20 | 13 |
| SPARTUN | YN | 13 | 4 | 9 | 0 |
| | FR | 35 | 0 | 35 | 0 |
| SPARTQA-HUMAN | YN | 29 | 20 | 6 | 3 |

Table 13: Analyzing wrong predictions in GT-PISTAQ. A: Missing/errors in **A**nnotation, C: rule-based **C**oreference issues in connecting extracted information, R: Shortcomings of the **R**easoner.

a grey car), Front(the car, a church), id 1= a cat, 2 = a grey car, the car, and 3 = a church. So we create new triplets in the form of BELOW(1, 2) and Front(2, 3).

To train the model, we pair each mention with its previous antecedent and use cross-entropy loss to penalize the model if the correct pair is not chosen. For singletons and starting mention of objects, the model should return class $0$, which is the $[CLS]$ token. Since the previous model does not support the plural antecedent (e.g., two circles), we include that by considering shared entity in pairs like both (two circles, the black circle) and (two circles, the blue circle) are true pairs.

As an instance of the importance of coreference resolution in spatial reasoning, consider this context "block A has one black and one green circle. The black circle is above a yellow square. The yellow square is to the right of the green circle. Which object in block A is to the left of a yellow square" The reasoner must know that the NTPPI(block A, one green circle) and RIGHT( the yellow square, the green circle) are talking about the same object to connect them via transitivity and find the answer.

**To evaluate** the coreference resolution module (Coref in Table 4), we compute the accuracy of the pairs predicted as Corefs. The Coref model achieves a high performance on all datasets. The performance is slightly lower on the SPARTQA-HUMAN dataset when SPARTUN is employed for additional pre-training. However, we observed that there are many errors in the annotations in SPARTQA-HUMAN, and the pre-trained model is, in fact, making more accurate predictions than what is reflected in the evaluation.

## C.3  Logic-based Spatial Reasoner

**To solely evaluate** the performance of the logic-based reasoner, we use the "GT-PISTAQ". We look into the errors of this model and categorize them based on the source of errors. The categories are

*missing/wrong ground truth direct annotations* (A), *rule-based Coreference Error* (C) in connecting the extracted information before passing to the reasoner, and *the low coverage of spatial concepts in the reasoner* (R). As is shown in Table 13, spatial Reasoner causes no errors for SPARTUN since the same reasoner has been used to generate it. However, the reasoner does not cover spatial properties of entities (e.g., right edge in "touching right edge") in SPARTQA and causes wrong predictions in those cases.

## D  SREQA on All Story Relations

| Datasets | F1 on SREQA |
|---|---|
| SPARTUN | 96.37 |
| SPARTQA-AUTO | 97.78 |
| SPARTQA-HUMAN | 23.79 |
| MSPRL (Used in RESQ) | 16.59 |

Table 14: The result of SREQA model only trained on all story relations of the SQA datasets.

Table 14 displays the results of the SREQA model trained and tested solely on all the story's relation extraction parts (step 1). During the evaluation, we also possess the same data preprocessing and gather annotations of all relations between stories' entities and select the best model based on performance on the development set.

Notably, the performance on the human-generated datasets, SPARTQA-HUMAN and RESQ, is significantly lower compared to the auto-generated datasets. As discussed in , the MSPRL datasets contain missed annotations, resulting in the omission of several relations from the stories' entities and incomplete training data for this phase. Similarly, the SPARTQA-HUMAN SPRL annotation, as discussed in Appendix C, exhibits some noise, particularly in coreference annotation, leading to similar issues as observed in MSPRL regarding annotation of all story relations.

Consequently, this reduced performance in all story relation extraction impacts the overall performance of the main SREQA model trained using two steps; however, as illustrated in the results of SREQA* in Table7, which utilizes Spartun instead of MSPRL for training on all story's relations, the performance substantially improves on the RESQ dataset.

## E  Large Language Models (LLMs)

Figure 11 presents examples showcasing predictions made by three Large Language Models (LLMs): GPT3.5-DaVinci, GPT4, and PaLM2, on a story from the SPARTQA-HUMAN dataset. These examples demonstrate that while these models, specifically GPT4 and PaLM2, excel in multi-hop reasoning tasks, solving spatial question answering remains a challenging endeavor.

To evaluate the LLMs' performance on spatial reasoning, we use $Zero\_shot$, $Few\_shot$, and $Few\_shot$+CoT. In the $Zero\_shot$ setting, the prompt given as input to the model is formatted as "Context: story. Question: question?" and the model returns the answer to the question.

In the $Few\_shot$ setting, we add two random examples from the training data with a story, all its questions, and their answers. Figure 12 depicts a prompt example for SPARTQA-HUMAN YN questions, passed to GPT3.5.

For $Few\_shot$+CoT, we use the same idea as (Wei et al., 2022) and manually write the reasoning steps for eight questions (from two random stories). The input then is formatted as "Context: story. Question: CoT. Answer. Asked Context: story. Question: question?". Figure 13 shows an example of these reasoning steps on RESQ.

### E.1  LLMs for Information Extraction

As discussed in Section 5.3.1, we utilize LLM, GPT3.5-Turbo, for information extraction from human-generated texts. The extraction process encompasses Entity, Relation, Relation Type, and coreference extractions from the story, as well as entity and relation extraction from the question. Additionally, LLM is employed to identify mentions of question entities within the text.

We construct multiple manually crafted prompt examples for each extraction task, as depicted in Figure 14. Subsequently, the extracted information is inputted into the reasoner module of PISTAQ to compute the answers.

In addition to our experiment, we attempted to incorporate LLMs as neural spatial reasoners but in a pipeline structure of extraction and reasoning. To do so, as illustrated in Figure 10, we add the extracted information of LLM with the written CoTs based on this extracted information to the prompt of a GPT3.5-DaVinci. The results, however, become even lower (62.62%) compared to GPT3.5-CoT with the main text (67.05%) when evaluated on

**Story:** a man in white shirt , black jacket , grey pants and black shoes is sitting on a wooden chair and talking on the phone . on the right a wooden bed with white bedcovers . on the left ( before the man ) a wooden desk and a vase with flowers . there is a black brief-case in front of the chair , and there is also a picture hanging on the wall above the bed .

**Relation_in_story**= [("a man", "in", "white shirt"), ("a man", "in", "black jacket"), ("a man", "in", "grey pants"), ("a man", "in", "black shoes"), ("A man", "sitting on", "a wooden chair"), ("a man", "talking on", "the phone"), ("a wooden bed", "on", "the right")("a wooden bed", "with", "white bedcovers"),("a wooden desk", "on", "the left"),("a wooden desk", "before", "the man"), ("a vase", "on", "the left"),("a vase", "before", "the man"),("a vase", "with", "flowers"),("a black brief-case", "in front of", "the chair"),("a picture", "hanging on", "the wall"),("a picture", "above", "the bed")]

**Question:** Is the man touching the chair? (the man, touching, the chair),

**Chain of reasoning:** ("A man", "sitting on", "a wooden chair") which is externally connected and above. The answer is Yes

**Question:** Is the bedcover below the picture?(the bedcover, below, the picture),

**Chain of reasoning:** ("a wooden bed", "with", "white bedcovers") which means bedcover has the same position as bedcover, ("a picture", "above", "the bed"), so (the picture, above, the bedcover), so (the bedcover, below, the picture). The answer is Yes.

Figure 10: We employ LLMs in both extraction and reasoning tasks, but in a disentangled manner. Initially, we extract information using LLMs, and subsequently incorporate this extracted information into the prompt alongside written CoTs based on the extracted data.

the RESQ dataset. This outcome highlights the superior ability of LLMs to capture information from natural language compared to structured data without fine-tuning.

| | | |
|---|---|---|
| SpaRTQA-Human | Story | There is a block called A. Inside there's a red oval that is above a blue triangle. The blue triangle is touching the right edge of A and the top of a large black square. Touching the bottom edge of A is a small orange square. The is a block called B Which is to the right and touching A. Inside B there are 5 triangles, a red one, a green one, a large yellow one, a brown one, and an orange one. The red and orange triangles are touching the left edge of B and the orange triangle is above the red triangle. The large yellow triangle is touching the top edge of B. The green triangle is touching the right edge of B. The brown triangle is below all of the other triangles and is touching the bottom of B. |
| | Question 1: | What is the relation between blue triangles and orange triangles? Answer: left, near |
| | Question 2: | Is the green thing above the yellow thing? Answer: No |
| | Question 3: | Does B have all of the triangles? Answer: No |
| GPT3.5 (Prediction) | Zero-shot | Q1: **Don't Know** 
 Q2: **No**, there is no green thing mentioned in the description. 
 Q3: **No** |
| | CoT | Q1:The blue triangle is touching the right edge of block A and the orange triangle is touching the left edge of block B. Block B is to the right and touching block A so the orange triangle is right of the blue triangle. The answer is [**Right**]. 
 Q2:The green triangle is to the right of the large yellow triangle and the large yellow triangle is touching the top edge of B. Using inverse rule, the green triangle is below the yellow triangle. The answer is **No**. 
 Q3: There are five triangles mentioned in block B, a red one, a green one, a large yellow one, a brown one, and an orange one. All of these triangles are in block B, so the answer is **Yes**. |
| GPT4 (Prediction) | Zero-shot | Q1: **Don't Know** 
 Q2: The d**escription does not provide enough information** to determine if the green triangle is above the yellow triangle. It only states that the green triangle is touching the right edge of block B and the large yellow triangle is touching the top edge of B. The relative vertical positions of the green and yellow triangles are not specified. 
 Q3: **Yes**, B has all of the triangles mentioned in the description. Inside block B, there are 5 triangles: a red one, a green one, a large yellow one, a brown one, and an orange one. |
| | CoT | Q1:The blue triangle is in block A, and the orange triangle is in block B. Block B is to the right and touching block A. So, the blue triangle is left of the orange triangle. The answer is [**Left**]. 
 Q2: The green triangle is touching the right edge of B, and the large yellow triangle is touching the top edge of B. It is not mentioned if the green triangle is above or below the yellow triangle. The answer is **Don't Know**. 
 Q3: There is a blue triangle in block A, and there are 5 triangles in block B. Since there is a triangle in block A, not all triangles are in block B. The answer is **No**. |
| PaLM2 (Prediction) | Zero-shot | Q1: The answer is **near**. The blue triangle and the orange triangle are both in the block B. The blue triangle is above the orange triangle. 
 Q2: **Yes**, the green thing is above the yellow thing. The green thing is a triangle that is touching the right edge of block B, and the yellow thing is a triangle that is touching the top edge of block B. 
 Q3: **Yes**, B has all of the triangles. The question asks if B has all of the triangles in the block. The answer is yes because the passage says that "Inside B there are 5 triangles, a red one, a green one, a large yellow one, a brown one, and an orange one." This means that B has all of the triangles in the block. |
| | CoT | Q1: The blue triangle is in block A and the orange triangle is in block B. Block A is left of block B so the blue triangle is left of the orange triangle. The answer is [**Left**]. 
 Q2: The green triangle is touching the right edge of B and the large yellow triangle is touching the top edge of B. The right edge of B is below the top edge of B. So the green triangle is below the yellow triangle. The answer is **No**. 
 Q3: There are 5 triangles in B. There is a red triangle in B. There is a green triangle in B. There is a large yellow triangle in B. There is a brown triangle in B. There is an orange triangle in B. The answer is **Yes**. |

Figure 11: An example of three Large Language Models (LLMs), GPT3.5 (davinci-text-003), GPT4, and PaLM2 prediction on SQA task evaluated with *zero_shot* and CoT prompting.

Context: There are three blocks called A, B, and C. A is to the right of B and B is to the right of C. In A, there is a large yellow square that is touching the right edge of A. Near and to the left of the square is a small black circle. There is a large black circle near and to the right of the small black circle. There is a medium blue circle below the large black circle. There is a large black triangle is below the medium blue circle. In B, there is a medium yellow square is touching the left corner of a small blue triangle. In C, there is a small black circle touching the bottom of C. There is a large yellow square near and above the small black circle. There is a large black triangle far above the large yellow square. There is a small yellow triangle near and to the left of the large black triangle. Near and to the left of the small yellow triangle is a small blue triangle.
Question: Is the small black thing in C below a large black thing in C? Answer: Yes
Question: Is the small blue thing in B to the right of a medium blue thing? Answer: No
Question: Is the medium yellow thing below a blue thing? Answer: No
Question: Is the small black in A near and to the right of a large yellow thing? Answer: Yes

Answer below questions:

Context: There is a block called A. In A there is a red triangle Which is above a black circle. The black circle is touching the bottom of A. A yellow square is touching the top edge of A and below that there is a red oval is touching the right edge of A. There is a second block call B Which is to the left of A. Inside B there are 3 squares, a brown one, a green one, and a red one. The green square is above the red square and touching the left edge of B. The brown square is touching both the bottom edge and the right edge of B. The red square is touching the left edge of the brown square.
Question: Is the oval to the right of the brown thing? Answer: No

🛈 Looking for ChatGPT?   Try it now ↗                                          ✕

Figure 12: Example of the input for GPT3.5($Few\_shot$). The $Zero\_shot$ setting is the same just do not have the first training example.

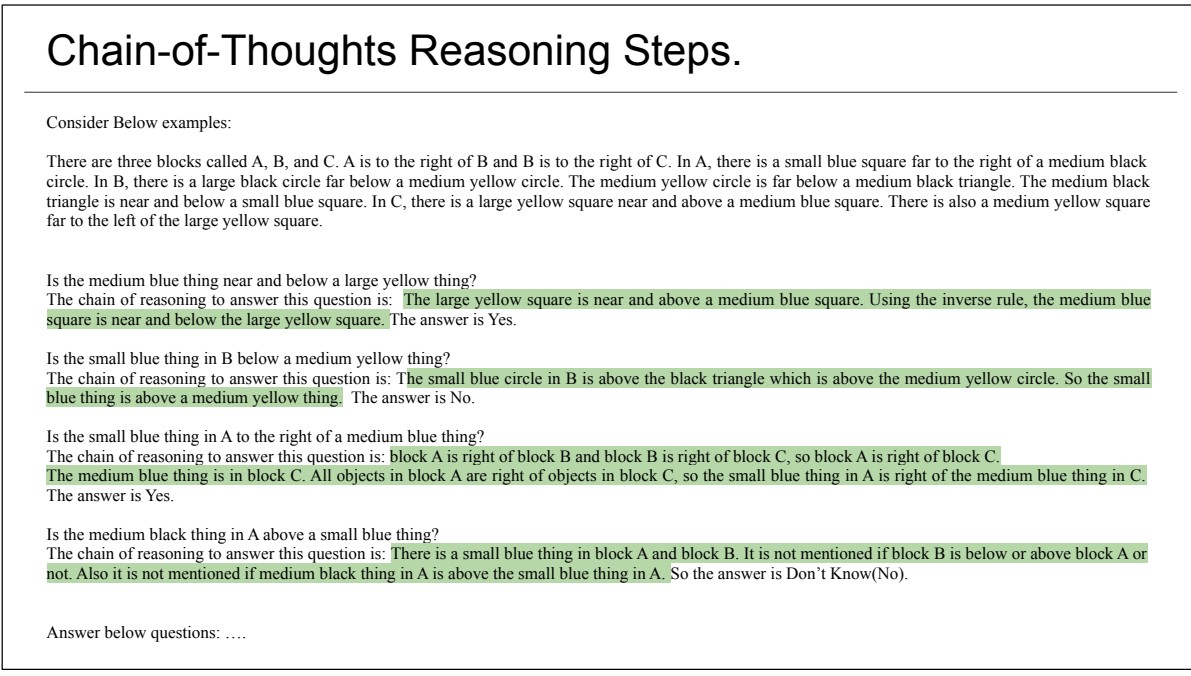

Figure 13: Example of the input for GPT3.5($Few\_shot$+Cot) with human-written Chain-of-Thoughts.

| Extraction | | Prompt Example |
|---|---|---|
| Story | Entity | Context 1: there are two social workers in the foreground . one wearing a red pullover and brown pants is bending over to access the blue paint . the other one in a red tee-shirt and black jeans is watching her . at the back of the room there is another worker wearing a white tee-shirt and blue jeans , acutally painting . there are many newspapers on the ground to protect the kindergarten floor . all three workers are wearing a mask . 

 entities = ['two social workers', 'the foreground', 'one', 'a red pullover', 'brown pants', 'the blue paint', 'the other one', 'a red tee-shirt', 'black jeans', 'her', 'the back', 'the room', 'another worker', 'a white tee-shirt', 'blue jeans', 'many newspapers', 'the ground', 'the kindergarten floor', 'three workers', 'a mask'] |
| | Relation | - a man in white shirt , black jacket , grey pants and black shoes is sitting on a wooden chair and talking on the phone. relation triplets: 
 [("a man", "in", "white shirt"), ("a man", "in", "black jacket"), ("a man", "in", "grey pants"), ("a man", "in", "black shoes"), ("A man", "sitting on", "a wooden chair"), ("a man", "talking on", "the phone")] 
 - on the right a wooden bed with white bedcovers. relation triplets: 
 [("a wooden bed", "on", "the right"), ("a wooden bed", "with", "white bedcovers")] |
| | Relation Type | If the relation set is: 
 Relation type set: 
 LEFT = to the left of another object, 
 DC= Disconnected, disconnected from other object, 

 (the wall, behind, the tourists), relation type is: ['BEHIND'] 
 (Lots of locals, in front of, a blue building) , relation type is: ['FRONT'] 
 (pictures, on, the wall), relation type is: ['FRONT', 'EC'] 
 (a clock, above, the blackboard), relation type is:['ABOVE'] |
| | Coreference | Context 1: Three women are sitting on a wooden bench in front of an about one metre high , red brick wall . they are all wearing skirts and jumpers ... 
 If the list of all entities is: 
 list_of_noun_phrases = ["three women", "they", "two of them", "a wooden bench", "an about one metre high red brick wall", "the wall", "skirts", ...] 
 The below list shows which noun phrases in the "list_of_noun_phrases" refers to which same phrase: 
 {"Three women": ["Three women", "they", "two of them"], 
 they: ["Three women", "they", "two of them"], 
 two of them: ["two of them"], 
 a wooden bench: ["a wooden bench",], .....} |
| Question | Relation and Type | If the relation set is: .... 

 Are the lamps behind the building?: list_of_dictionary = [{"triplet": ("the lamps", "behind", "the building"), "relation type": ['BEHIND']}] 
 Is the camera in front of the all kids?: list_of_dictionary = [{"triplet": ("the camera", "in front of", "the all kids"), "relation type": ['FRONT']}] 
 Is a flag to the left of the stairs?: list_of_dictionary = [{"triplet": ("a flag", "to the left of", "the stairs"), "relation type": ['LEFT']}] |
| Question entity to Story Mentions | | This should consider the exact or partially matching based on the phrase root. 
 For examples,"{0: "small window", 2: "large window", 5: "three windows"}" all can be matched with "the window" since the root is window here. 
 Also the output should be in the form of only a python dictionary like {"the window": [0,2,5]}. |

Figure 14: The example of prompts used for LLMs (GPT3.5-Turbo) in information extraction.