# OpenReview forum: "Disentangling Extraction and Reasoning in Multi-hop Spatial Reasoning"
_EMNLP/2023/Conference — EMNLP 2023 Findings_

### Official Review · Reviewer_6kdH · 2023-08-02

**Soundness:** 3

**Excitement:**

3: Ambivalent: It has merits (e.g., it reports state-of-the-art results, the idea is nice), but there are key weaknesses (e.g., it describes incremental work), and it can significantly benefit from another round of revision. However, I won't object to accepting it if my co-reviewers champion it.

**Paper Topic And Main Contributions:**

This paper focuses on spatial role extraction and spatial relation extraction. Its main contribution is to disentange the processes of spatial relation extraction to different stages. In details, it proposes three different models: a pipeline of extraction and symbolic reasoning, an end-to-end PLM in a QA format, and  an end-to-end neural model with explicit layers of extraction and reasoning. The experimental results on multiple datasets show the effectiveness of the proposed models.

**Questions For The Authors:**

1)  PISTAQ is better than SREQA on SPART datasets, However, it performs worse than SREQA on RESQ. Why?

2) The performance of entity coreference is still poor and most models are less than 70 in F1-score. Why can the accuracy of the proposed Coref module acheive 99? How about the Precision and Recall?

**Reasons To Accept:**

1) This paper proposed three different model on spatial relation extraction.

2) The proposed models achieve good performance on multiple datasets.

3) This paper is well-written.

**Reasons To Reject:**

1) I mainly concern its novelty. The three proposed models are not new, because pipeline models and QA-style models are widely used in IE field.

2) Which model is the best for a IE task? Pipeline model? Unified model? Joint model? or multitask model? I think this paper should compare   the proposed models with other framework, e.g., joint or multitask framework.

3) The baselines are weak. This paper only compared the proposed models with some basic models (e.g., Majority, BERT,GPT3) and did not compare them with the SOTA models.

4) PISTAQ is better than SREQA on SPART datasets, However, it performs worse than SREQA on RESQ. I would like to see the reason.

5) The motivation is not clear. This paper did not answer the question: Why is disentangling extraction and reasoning better than joint  extraction and reasoning in multi-hop spatial reasoning?

**Reproducibility:**

4: Could mostly reproduce the results, but there may be some variation because of sample variance or minor variations in their interpretation of the protocol or method.

**Reviewer Confidence:**

5: Positive that my evaluation is correct. I read the paper very carefully and I am very familiar with related work.

---

> ### Author Rebuttal · Authors · 2023-08-28
>
> We appreciate your positive feedback and recognizing and acknowledging the contributions of our paper. We believe that your comments and questions can effectively improve the quality and understanding of our paper.
>
> Please find below our responses addressing the points and questions you have raised.
>
> - Reason to reject 1: The main idea of this paper is to propose the different ways of disentangling extraction and rule-based reasoning as a pipeline and an end-to-end solution. These models are proposed for the multi-hop reasoning task, which is more challenging than a simple IE task, which extracts direct information and needs complex logical computation.
> In the related work section 2, we mentioned the previous research in this area and their differences. Several end-to-end models for SQA tasks exist, and PLMs are the SOTA in this area. Even compared to more complex models (lines 108-117). As a result, we select PLMs as the primary end-to-end baselines to compare our model results.
> On the other hand, the previous pipeline models, with some separate extraction and reasoning, mainly use logical rules to keep the consistency of the extraction prediction (lines 132-135). Also, other works use simple summation over the computed probability of extraction modules instead of logical rules (lines 135-138) for the reasoning module. So, using the implemented crisp logic in the reasoning part to solve the multi-hop reasoning in both forms that we have, PistaQ as a separate module or as in SREQA in data preprocessing was not discussed before.
>
> - Reason to reject 2: Just to confirm again, we do not propose methods for IE but for multi-hop question-answering task. As we state in several places in the paper, such as in the introduction (lines 87 - 104) and conclusion(lines 560-577), our experiments show that PISTAQ (pipeline) is better than other models in controlled environments.  The complexity of these datasets is more related to conducting several reasoning steps and demands accurate, logical computations where a rule-based reasoner excels and results in high performance. However, in real-world data, these implemented logical rules have some limitations. The extraction module is weaker due to the fewer annotations, so using SREQA (End-to-End modeling + logical pre-processing) works better. The end-to-end structure of SREQA also helps to capture some spatial commonsense knowledge from the training data alongside utilizing the benefit of the rule-based reasoner in the data preprocessing. Compared to these, if we use LLMs for extraction in PistaQ, we can have more accurate extracted information from even real-world datasets, which are more complex. Of 25 sampled questions from RESQ, only 7 can be solved without relying on spatial commonsense information (lines 451-459). The PISTAQ model using LLM as extraction modules can solve all 7 questions.
>
>    We have conducted comparisons against recent SOTA, especially LLMs, on specific datasets relevant to our research scope. The current LLMs surpass all previous unified, joint, and multitask SOTAs[1,2,3] and as shown in [4], using SPARTUN as extra supervision in BERT already surpassed the models with more complex structures. We have some related research in section 2, which shows how our model captures some ideas from prior work or how it differs from theirs. If you are aware of any relevant work that could serve as additional baselines for our study, we would greatly appreciate your insights to help enhance the comprehensiveness of our analysis.
>
>
> - Reason to reject 3: The baselines we compared our proposal against are the state-of-the-art performing models published recently in top NLP venues. Consider that this is a relatively new task and does not have many baselines. Besides, the current LLMs, specifically LLMs with Chain-of-out, already surpassed other models[1], specifically in the reasoning area. Therefore, the baselines are strong, and the SOTA on the task.
>
> - Reason to reject 4 and Question 1: Thank you for the good question. We make sure to include this discussion in the paper. The SREQA performance does not surpass PISTAQ’s in a controlled environment. The complexity of these datasets is more related to conducting several reasoning steps and demands accurate, logical computations areas where a rule-based reasoner excels. So, in a controlled environment, the result of PistaQ with a reasoner module is higher than any other model. The reason for the lower performance of  PISTAQ on ReSQ is discussed in lines 451-464 and the better result of SREQA on ReSQ is discussed in lines 483-489.
>
>
> - Reason to reject 5: Throughout our work, we emphasize our motivation, which centers on evaluating the efficacy of disentangling extraction and reasoning within the context of multi-hop spatial reasoning. Our motivation stems from the limitations of existing black-box PLMs in lines 35-45 (the performance of PLMs is not consistent and mostly captures the pattern[5]), the benefits of using extracted information (lines 46-53), and human performance(line 54).
>
>    Many works, including the cited paper in our work and our experiments, show that the black-box NN models, even the powerful LLMs, have limitations in multi-step reasoning. Hence, in our disentangled model, we utilize NN models for extraction instead of the whole reasoning process. Subsequently, we perform rule-based reasoning over them to prevent erroneous reasoning computation. As a result, our disentangled model performs better than joint models.
>
> - Question 2:  Firstly, the coreference challenges in the synthetic datasets (Spartun and SpartQA-Auto) are not as challenging as datasets on the Coreference resolution task. Also, these datasets contain large training sets, which helps the learning process to effectively capture the coreferences within this context. Secondly, to consider the plural coreference, we evaluate the models based on the accuracy of correctly predicting coreference pairs (as detailed in Appendix C2, Line 911). Moreover, ignoring the plural coreferences, we additionally evaluated the model by the B3 score, which still indicated the high performance of ~99% B3 on synthetic benchmarks. As an example, below, you can see the result of our model ignoring plural coreference on SPARTUN:
> Test Precision B3:  0.99005
> Test Recall B3:  0.99217
> Test Macro-F1 B3:  0.99107
> Test accuracy:  0.98741
>
>
> References:
>
> [1]Wei, Jason, Xuezhi Wang, Dale Schuurmans, Maarten Bosma, Fei Xia, Ed Chi, Quoc V. Le, and Denny Zhou. "Chain-of-thought prompting elicits reasoning in large language models." Advances in Neural Information Processing Systems 35 (2022): 24824-24837.
>
> [2]Ozturkler, Batu, Nikolay Malkin, Zhen Wang, and Nebojsa Jojic. "ThinkSum: Probabilistic reasoning over sets using large language models." arXiv preprint arXiv:2210.01293 (2022).
>
> [3]Bang, Yejin, Samuel Cahyawijaya, Nayeon Lee, Wenliang Dai, Dan Su, Bryan Wilie, Holy Lovenia et al. "A multitask, multilingual, multimodal evaluation of chatgpt on reasoning, hallucination, and interactivity." arXiv preprint arXiv:2302.04023 (2023).
>
> [4]Roshanak Mirzaee and Parisa Kordjamshidi. 2022. Transfer Learning with Synthetic Corpora for Spatial Role Labeling and Reasoning. In Proceedings of the 2022 Conference on Empirical Methods in Natural Language Processing, pages 6148–6165, Abu Dhabi, United Arab Emirates. Association for Computational Linguistics.
>
> [5]Roshanak Mirzaee, Hossein Rajaby Faghihi, Qiang Ning, and Parisa Kordjamshidi. 2021. SPARTQA: A Textual Question Answering Benchmark for Spatial Reasoning. In Proceedings of the 2021 Conference of the North American Chapter of the Association for Computational Linguistics: Human Language Technologies, pages 4582–4598, Online. Association for Computational Linguistics.

---

### Official Review · Reviewer_WRQX · 2023-08-04

**Soundness:** 3

**Excitement:**

4: Strong: This paper deepens the understanding of some phenomenon or lowers the barriers to an existing research direction.

**Paper Topic And Main Contributions:**

This paper proposes to disentangle the information extraction process with reasoning process for multihop spatial reasoning dataset, and design both symbolic pipeline disentangling and end-to-end neural disentangling to achieve the objective. To showcase the effect of the disentanglement, the authors compare with PLMs such as BERT and GPT-3 which produce direct inference given the story and question.

Extensive experiments are conducted over different datasets, automatic or human-authored, with or without SPRL labels, to empirically verify the effect of different components, and analyze the models' generalization towards unseen datasets during training.

**Questions For The Authors:**

Refer to the above.

**Reasons To Accept:**

1. The paper raises an interesting and valuable problem of whether disentangling the extraction and reasoning process leads to improved spatial reasoning.
2. Comprehensive model designs and comparisons are given to ablate specific functionalities with empirical results, including direct black-box inference, neural disentanglement, and symbolic disentanglement.
3. The results indicate the potential of using specific reasoning models to replace LLMs in reasoning-intensive tasks.

**Reasons To Reject:**

1. The descriptions are intensive, but lack clearer organizations which makes it a bit challenging to follow. A running example could be added to make description much easier.
2. The settings of different proposed models are not so clear, as well as some of the illustrations. For example:
- What are logic rules used in the prolog for the reasoning part?
- Is BERT-EQ essentially similar to BERT but with data augmentations using SPRL labels?
- For SREQA, how is entity selection  (via BIO, similar to PISTAQ?) and the final answer produced given the predicted pairs?
3. The proposed models still require SPRL labels for training the extraction module. The authors mention that it is beneficial to use LLMs to produce zero-shot or few-shot extractions, but is there any empirical results for the final performance?

**Reproducibility:**

4: Could mostly reproduce the results, but there may be some variation because of sample variance or minor variations in their interpretation of the protocol or method.

**Reviewer Confidence:**

4: Quite sure. I tried to check the important points carefully. It's unlikely, though conceivable, that I missed something that should affect my ratings.

---

> ### Author Rebuttal · Authors · 2023-08-28
>
> We appreciate the valuable points and comments, which will significantly enhance the paper's readability and clarity. Please find below our responses addressing the specific points and questions you have raised.
>
> - Reason to reject 1: Thank you for raising this concern. We have indeed incorporated running examples in Figures 2 and 3 with the main models. However, considering the extra page in the camera-ready version, we will further elaborate on these examples in the context of the paper to enhance clarity, especially for the SREQA model.
>
> - Reason to reject 2-a (Logical rules): The logic rules employed for reasoning are consistent with those outlined in the referenced paper (line 223), and these rules are also documented in Table 12. We will ensure to clear this in the text with reference to Table 12.
>
> - Reason to reject 2-b (BERT-EQ): Indeed. In lines 74- 75 and 234, we mention extra questions, meaning extra than the primary question. This also is illustrated in Table 2 with QA(main questions)+SPRL(augmented questions). We also used similar naming with extra EQ in the end for the same reason. We will further address this in section 3.2 to enhance clarity.
>
> - Reason to reject 2-c (SREQA): In response to your question about entity selection and final answer production in SREQA, please note that Figure 3.b and lines 286-287 show that the entity selection data comes directly from the annotation in the first training phase. In the next phase, we employ the same trained extraction models as utilized in PistaQ(lines 296-297). We make sure to clear this in the text. The final answers come from the output of each relation classifier, which identifies the relations between entity pairs. These details are more elaborated in lines 301-306 of the paper.
>
> - Reason to reject 3: We do not have the empirical result on the whole ReSQ datasets since structuring the extracted information needed engineering and was time-consuming. So we only showed some case-studies of this idea in the paper in subsection 5.3. However, we have the result on a subset of ReSQ. Of 25 sampled questions only 7 of them can be solved without relying on spatial commonsense information (more details lines 451-459). The PISTAQ model using LLM as extraction modules can solve all 7 questions. We will add this result in the next revision.

---

### Official Review · Reviewer_K1u1 · 2023-08-04

**Typos Grammar Style And Presentation Improvements:** 1.	In Table 3, dataset MSPRL is menti…
**Soundness:** 3

**Excitement:**

3: Ambivalent: It has merits (e.g., it reports state-of-the-art results, the idea is nice), but there are key weaknesses (e.g., it describes incremental work), and it can significantly benefit from another round of revision. However, I won't object to accepting it if my co-reviewers champion it.

**Paper Topic And Main Contributions:**

To show benefits of segregating the processes of information extraction and reasoning in the task of spatial question answering, the paper proposes 3 models: PistaQ, BERT-EQ and SREQA. In brief, PistaQ is a model based on extracting relations and then perform symbolic reasoning. BERT-EQ is an end-to-end pre-trained language model that uses the same spatial information supervision but in question-answer format. Lastly, SREQA model is again an end-to-end neural model with explicit layers of extraction and reasoning. Various experiments are conducted over 3 datasets: SPARTQA, SPARTUN and RESQ, which show the efficacy of separating the processes information extraction and reasoning.

**Reasons To Accept:**

1.	This paper demonstrates the effectiveness of separating the process of information extraction from reasoning while performing spatial question answering.
2.	Although the quantitative and qualitative are strong against the baseline models, yet, they may lead a reader to a dilemma on which model, PISTA-Q or SREQA, to use for any use- case.

**Reasons To Reject:**

1.	The structure of the paper is not good. For instance,
      a.	Text below section 3 should briefly introduce all the 3 models (with names of models) and should refer to subsections 3.1, 3.2 and 3.3 to provide the reader an idea of what to expect.
      b.	Names of the models should be easy to remember. For example, what is the purpose of EQ in BERT-EQ. As a reader, I don’t understand it until I read section 3.2.
      c.	Heading of Section 3.1 should be PISTAQ: ….. to make it evident that the section talks about first model.
      d.	Heading of Section 3.2 should be BERT-EQ: ….. to make it evident that the section talks about second model.
      e.	Heading of Section 3.3 should be SREQA: ….. to make it evident that the section talks about second model.
      f.	Is the section starting at Line 525 a subsection or something different?

2.	Table 2 does not help in dissecting the attributes of proposed models among each other and with the baselines. The authors are requested to make it easy for the readers to get an understanding of the various models. For instance, a suggestion is to make a table where every row is a model name and columns are 4 or 5 attributes which the authors feel critical to distinguish the various models. This will help in the ablation analysis aswell. Moreover, current Table 2 does not contain baselines: Majority Baseline and GT-PISTAQ. The purpose of this table is to give readers a quick glance about all the models discussed in the paper with their attributes.

3.	Sections discussing results simply report the results without any interesting insights. My suggestion will be to discuss the results with respect to the new Table 2 mentioned in point 2. This will keep readers interested in reading the results with some insights about attributes of the proposed models.

4.	The paper uses too many acronyms, and some acronyms have an unusual mention. For eg. SPARTUN where size of “S” is smaller than “RTUN” but bigger than “PA”.

5.	The authors are requested to create separate sections for quantitative results which can have Table 3-6 at one place and a qualitative section which can have Figures 4 and 5.

**Reproducibility:**

3: Could reproduce the results with some difficulty. The settings of parameters are underspecified or subjectively determined; the training/evaluation data are not widely available.

**Reviewer Confidence:**

4: Quite sure. I tried to check the important points carefully. It's unlikely, though conceivable, that I missed something that should affect my ratings.

---

> ### Author Rebuttal · Authors · 2023-08-28
>
> We appreciate the valuable suggestions you've provided for improving our paper. We will certainly take these into account as we work on revising the paper.
>
> It's important to note that the suggestions regarding section organization and formatting can be helpful for enhancing the paper's structure and readability. Still, we acknowledge that these aspects alone do not constitute grounds for rejection.
> Regarding soundness scoring, “ Some of the main claims/arguments are not sufficiently supported, there are major technical/methodological problems.”, we would greatly appreciate further clarification on which specific claims may not have been adequately supported or which technical/methodological aspects are of concern. This information would be tremendously helpful as we work towards addressing any potential shortcomings and ensuring the overall quality of the paper.
>
> Please find below our responses addressing the points and questions you have raised.
>
>  - Regarding "dilemma on which model, PISTA-Q or SREQA, to use for any use- case":
>
>    As we state in several places in the paper, such as in the introduction (lines 87 - 104) and conclusion(lines 560-577), our experiments show that PISTAQ is better than other models in controlled environments.  The complexity of these datasets is more related to conducting several reasoning steps and demands accurate, logical computations where a rule-based reasoner excels compared to black-box PLMs and results in higher performance.
>
>    However, on real-world data, these implemented logical rules has some limitation(not considering spatial commonsense), and also, the extraction module is weaker due to the fewer annotations, so using SREQA works better. The end-to-end structure of SREQA also helps to capture some spatial commonsense knowledge from the training data alongside using the benefit of the rule-based reasoner in the data preprocessing.
>
>    Compared to these, if we use LLMs for extraction, we can have more accurate extracted information from even real-world datasets which are more complex. In our case study (section 5.3), our model effectively answered all seven out of twenty-five sampled ResQA questions, which can be solved without relying on spatial commonsense information.
>
> - Reason to Reject1: (a,c,e,d) We prefer to have a short description of the model in the subsections of section 3 as the models' names are not described before. (b) The first time we mention the name of BERT-EQ is in subsection 3.2 when it is discussed. (f) Line 525 is part of the 5.3 section on LLM on spatial reasoning and is a title for the following paragraph (the “:” is missed after that). We can add another subsection there to clear that.
>
> - Reason to Reject2: Although your suggestion might be another way to list the models, we made a table to clarify the source of their supervision (not a glance of all models), as this is the most important feature. The rest of the information is available in the text when we describe the models. Also, as we described models in Subsection 4.2, the Majority baseline and GT-PistaQ doesn’t have any supervision, while as the caption state, Table 2 is a training supervision of baselines in this paper.
>
> - Reason to Reject3: Thank you for your suggestion. However, we chose the current structure to represent our results. The result section includes the report numbers and proper discussions and further information on the rationale behind each model’s performance. We respectfully disagree that the current section does not provide interesting discussions.
>
> - Reason to Reject4: All the acronyms are described when they are mentioned for the first time. Also, we consider most of them to be well-known acronyms in the NLP. Please let us know if there is a specific acronym that we did not describe, and we will ensure to include that given the extra page. Regarding the specified dataset name, we have only used the dataset name which was proposed in prior research and is not our own invention.
>
> - Typo 1: MSPRL is the backbone of the ReSQ dataset, and we described it in lines 329-337.
>
> - Typo 3: We use Figure in the text, but in the parentheses, we use Fig as a short form.
>
> - Typo 4.a: We don’t have the coreference resolution in question as we describe it in the text (lines 217-221). To connect question entities to the stories, we use Rule-Coref, which we also have in Figure 2. We make this clear in the text.
>
> - Typo 4.b: All of the computation of spatial reasoner happens inside the prolog considering all the implemented rules shown in Table 12. Overall, Prolog would get the question as a query and the relations as facts; it will return an answer to our query directly. We will ensure to extend the information related to the spatial reasoner in text and within the figure, enabling a more coherent visual representation of its processes.

---

### Meta-Review · Area_Chair_WstE · 2023-09-08

**Recommendation:** 3

**Metareview:**

The authors propose an approach for spatial reasoning over text to tackle Spatial Question Answering. The proposed solution disentangles the processes of information extraction and reasoning, and this is demonstrated to be effective on multiple Spatial QA datasets.
The paper has some merits, especially the experimental section which is very exhaustive. Still the paper can be largely improved, in particular it can be better organized and made more clear.

Furthermore, the authors proposed three different solutions (a pipeline of linguistic modules, end two end-to-end systems based on pre-trained Language Models) which are tested only on 3 different datasets resulting in no clear best system (depending on the dataset the best solution changes). This is very problematic since it is hard to draw conclusions. What model should a practitioner use? I recommend the authors to try to take the best from each proposed solution and verify whether they can come up with individual model that works well in every setting.

---

### Decision · Program_Chairs · 2023-10-07

**Decision:**

Accept-Findings

**Comment:**

The authors propose an approach for spatial reasoning over text to tackle Spatial Question Answering. The proposed solution disentangles the processes of information extraction and reasoning, and this is demonstrated to be effective on multiple Spatial QA datasets.
The paper has some merits, especially the experimental section which is very exhaustive. Still the paper can be largely improved, in particular it can be better organized and made more clear.

Furthermore, the authors proposed three different solutions (a pipeline of linguistic modules, end two end-to-end systems based on pre-trained Language Models) which are tested only on 3 different datasets resulting in no clear best system (depending on the dataset the best solution changes). This is very problematic since it is hard to draw conclusions. What model should a practitioner use? I recommend the authors to try to take the best from each proposed solution and verify whether they can come up with individual model that works well in every setting.